# All Roads Lead to Cathepsins: The Role of Cathepsins in Non-Alcoholic Steatohepatitis-Induced Hepatocellular Carcinoma

**DOI:** 10.3390/biomedicines10102351

**Published:** 2022-09-21

**Authors:** Hester van Mourik, Mengying Li, Sabine Baumgartner, Jan Theys, Ronit Shiri-Sverdlov

**Affiliations:** 1Department of Genetics and Cell Biology, NUTRIM—School for Nutrition and Translational Research in Metabolism, Maastricht University, 6229 ER Maastricht, The Netherlands; 2Department of Precision Medicine, GROW—School for Oncology and Reproduction, Maastricht University, 6229 ER Maastricht, The Netherlands; 3Department Nutrition and Movement Sciences, NUTRIM—School for Nutrition and Translational Research in Metabolism, Maastricht University, 6229 ER Maastricht, The Netherlands

**Keywords:** NASH, HCC, cathepsins, cancer hallmarks

## Abstract

Cathepsins are lysosomal proteases that are essential to maintain cellular physiological homeostasis and are involved in multiple processes, such as immune and energy regulation. Predominantly, cathepsins reside in the lysosomal compartment; however, they can also be secreted by cells and enter the extracellular space. Extracellular cathepsins have been linked to several pathologies, including non-alcoholic steatohepatitis (NASH) and hepatocellular carcinoma (HCC). NASH is an increasingly important risk factor for the development of HCC, which is the third leading cause of cancer-related deaths and poses a great medical and economic burden. While information regarding the involvement of cathepsins in NASH-induced HCC (NASH-HCC) is limited, data to support the role of cathepsins in either NASH or HCC is accumulating. Since cathepsins play a role in both NASH and HCC, it is likely that the role of cathepsins is more significant in NASH-HCC compared to HCC derived from other etiologies. In the current review, we provide an overview on the available data regarding cathepsins in NASH and HCC, argue that cathepsins play a key role in the transition from NASH to HCC, and shed light on therapeutic options in this context.

## 1. Introduction

The term “cathepsin” (derived from the Greek word “Καθεψίνη”, which means to “boil down” or “digest”) was first coined in 1929 by Nobel Prize winner Richard Willstätter to describe all proteases that degrade proteins [1]. Cathepsins are now recognized as proteases that are predominantly active in the endo-lysosomal compartment [2]. These proteases are subdivided into three distinguished families based on which amino acid is present at their active site, i.e., the serine proteases (cathepsin A, G), the aspartic proteases (cathepsin D, E), and the cysteine proteases (cathepsin B, C, F, H, K, L, O, S, V, X, W) [3]. The most abundant cathepsins are cathepsin B, D, and L (CTSB, CTSD, CTSL respectively) [4]. Although most cathepsins are ubiquitously expressed throughout the body, some show tissue-specific expression [5]. This tissue-specific expression is important since the function of cathepsin can differ per tissue type.

Cathepsins execute diverse functions during homeostasis in which they cleave proteins within essential processes such as the immune response and energy metabolism [6]. They are expressed as inactive precursors and are activated through proteolytic cleavage of the N-terminal pro-peptide [7,8]. For optimal activity, cathepsins require an acidic pH, which can be found inside the endo-lysosomal compartments [3,5]. However, several cathepsins are also stable in more neutral pH conditions outside of their endo-lysosomal compartment and even outside of the cell [9]. While extracellular cathepsins have physiological functions in processes such as wound healing, they are mainly associated with pathological conditions [2,9], including several metabolic diseases and various types of cancer [3,10], where increased cathepsin concentrations have been associated with a poor prognosis [11,12]. Among others, the pathogenic role of cathepsins was demonstrated in hepatocellular carcinoma (HCC) [11,13]. In addition, cathepsins were found to participate in the disease progression of non-alcoholic steatohepatitis (NASH), an increasingly important risk factor for the development of HCC [14,15]. The role of cathepsins in NASH-induced HCC (NASH-HCC), however, has not been elucidated yet. Since NASH-HCC is a condition with limited therapeutic options and taking into account the involvement of cathepsins in both processes, targeting specific cathepsins might be a viable therapeutic option.

In this review, we provide an overview of the role of cathepsins in NASH and HCC and argue that cathepsins play a key role in the transition from NASH to HCC. First, we will describe the prevalence and disease characteristics of NASH-HCC. Subsequently, we will discuss the function of the most studied cathepsins (CTSB, CTSD, CTSL, and cathepsin (CTS)S) in NASH, followed by a description of the mechanisms of cathepsins in HCC by means of the renowned “hallmarks of cancer” [16,17,18]. Lastly, the therapeutic potential of targeting cathepsins in HCC and specifically NASH-HCC will be discussed.

## 2. NASH-HCC Is a Global Health Problem

Worldwide, the prevalence of non-alcoholic fatty liver disease (NAFLD) in the adult population is 25% [19] and is expected to rise to more than 30% in the next decade [20]. NALFD is defined by excessive hepatic steatosis (≥5% in hepatocytes), not caused by excessive alcohol ingestion or by other chronic hepatic diseases such as viral hepatitis [21,22]. Furthermore, the disease is closely associated to conditions such as metabolic syndrome and type 2 diabetes [22]. NAFLD patients have an increased risk of developing liver-related morbidity and mortality [23]. In addition, NAFLD patients are at high risk of developing cardiovascular disease, chronic kidney disease, and extrahepatic malignancies (e.g., colorectal, uterine, and breast cancer), among others [24,25]. For about 30% of the patients affected by NALFD, the disease can progress into the more severe non-alcoholic steatohepatitis (NASH) [19,26]. Besides hepatic fat accumulation, NASH entails lobular inflammation and injury to the hepatocytes [15]. In addition, the liver of these patients can become fibrotic, making them at risk of the development of liver cirrhosis (5–18% for NASH patients without fibrosis, up to 38% for NASH patients with fibrosis [26]) and hepatocellular carcinoma (HCC) (2.4–12.8% for cirrhotic NASH patients [27]) [28,29].

HCC is the most common type of primary liver cancer [30]. Furthermore, it is the third leading cause of cancer-related deaths worldwide [31]. Currently, NASH is the most emergently growing cause for the development of HCC in patients that receive liver transplantations or that are on the waiting list [32,33]. A subset of patients develops cirrhosis as a result of NASH. These cirrhotic patients are at risk for developing HCC and they enter a surveillance program in order to detect the tumor on time [34,35]. However, the tumor is missed in one-fifth of these cirrhotic patients during these screening sessions, resulting in late-stage diagnosis of HCC [36]. In addition, it is now acknowledged that NASH patients without the presence of cirrhosis can also develop HCC [37,38,39]. In the absence of cirrhosis, these patients do not enter the surveillance program, which often results in a late-stage diagnosis as well. As curative treatment options require the cancer to be diagnosed in an early stage, many patients therefore miss this opportunity and are subjected to systemic treatment or palliative care [40]. As such, HCC poses a great burden on health and the economy [41].

CTSD has been found to be elevated in the plasma of patients with NASH compared to healthy individuals [14]. In addition, CTSB, CTSD, CTSL, and CTSS were found to be increased in HCC patients and associated with poor prognosis [11,42,43]. Ample studies have been performed regarding the role of cathepsins in several types of cancers. In the next section, the mechanism of these cathepsins in NASH and HCC will be addressed.

## 3. Mechanisms of Cathepsins in NAFLD/NASH

### 3.1. Cathepsins Induce Apoptosis in NASH

Apoptosis is a critical process in the disease progression of NASH and other liver diseases [44]. Apoptosis is mediated through extrinsic or intrinsic factors (classical pathway) that lead to the activation of effector caspases, resulting in apoptosis [45]. Next to the classical pathway, apoptosis can also be triggered alternatively through lysosomal disruption, which leads to the release of lysosomal proteases into the cytosol. It is generally accepted that cathepsins, subsequent to lysosomal membrane permeabilization (LMP), enter the cytosol to induce the apoptotic machinery by cleaving several proteins involved in the cascade [46]. In NALFD patients, disease severity has been associated with lysosomal permeability, and therefore the release of cathepsins into the cytosol [47]. An imbalance in the rate of cell proliferation and apoptosis is also thought to contribute to the NASH pathology, since an excess of apoptosis results in increased rates of regeneration and damage [47].

Most of the evidence related to cathepsin-induced cell death in liver diseases originates from studies on HCC. We will therefore provide more detailed insights on the role of cathepsins in apoptosis in the section on HCC.

### 3.2. Cathepsins Play a Role in the Disease Progression of Conditions with Lipid Accumulation

Cathepsins play a role in the pathology of several diseases related to lipid accumulation, such as metabolic syndrome and NAFLD/NASH. Lipids that accumulate in tissues other than adipose tissue enter the cells and build up in the lysosomes, which results in the release of cathepsins and other lysosomal enzymes. This process culminates in cell death, also known as lipotoxicity [42]. Therefore, cathepsins are thought to be involved in conditions where excess lipid accumulation plays a role.

Many patients that suffer from metabolic syndrome and NAFLD present with insulin resistance, which increases the risk of the development of type 2 diabetes (T2D) [48]. Cathepsins play a role in insulin sensitivity as well. Insulin sensitivity, CTSD levels, and activity were measured in overweight and obese individuals [49]. The activity of plasma CTSD was inversely associated with the whole-body sensitivity of insulin, i.e., the more likely it is that these overweight/obese individuals will develop type 2 diabetes (T2D), the higher the activity of their plasma CTSD. Additionally, in T2D patients, plasma CTSD activity appeared to be significantly higher compared to healthy individuals [50]. In addition, T2D markers (plasma glucose, HbA1c) were significantly correlated to plasma CTSD activity. Moreover, CTSS levels were higher in a population of T2D patients that had cardiovascular disease compared to T2D patients that did not have cardiovascular disease and the plasma CTSS levels were positively associated to components of the metabolic syndrome in obese adults [51,52].

In addition to insulin resistance, cathepsins have been linked to the deregulation of the liver metabolism in NAFLD and NASH. Apart from the observation that lipid accumulation in the liver and muscle in NALFD patients is positively associated with the levels of CTSD [53], multiple studies also point towards the involvement of other cathepsins in the more progressed form of NALFD, NASH. Increased levels of CTSD were observed in NASH patients compared to healthy individuals [14]. Furthermore, the involvement of cathepsins in NASH was also demonstrated in mice. Wild-type (WT) mice that were fed a fructose-palmitate-cholesterol (FPC) diet developed NASH within 18 weeks. Fang et al. demonstrated that these NASH mice showed elevated levels of CTSB compared to mice on a low-fat diet [54]. In addition, they showed that *CTSB* knockout mice on an FPC diet had lower levels of plasma total cholesterol, lower levels of liver triglyceride, and higher levels of plasma HDL compared to WT mice on an FPC diet. In *CTSB* knockout mice, decreasing inflammatory cytokine and chemokine levels were also observed, resulting in lower inflammation [54]. Improved liver function upon CTSB inhibition was also observed in NASH mice that were fed a methionine choline diet (MCD) compared to MCD-fed mice without the CTSB inhibitor [55].

In line, inhibition of CTSD in a model of murine NASH (*LDLR*^−/−^ mice on a high-fat, high-cholesterol (HFC) diet) resulted in decreased levels of plasma cholesterol and triglycerides, in addition to a decrease in hepatic cholesterol and total triglycerides compared to NASH mice in which CTSD was not inhibited. Additionally, genes related to hepatic lipid metabolism, such as *PPAR-γ*, were decreased in expression upon CTSD inhibition [56]. More specifically, when the extracellular fraction of CTSD was inhibited in HFC *LDLR*^−/−^ mice, the hepatic total cholesterol and triglycerides were significantly decreased. Furthermore, when the intracellular CTSD inhibitor was compared to the extracellular CTSD inhibitor, the hepatic total triglyceride levels were lower and the levels of fecal bile acids (indicating increased cholesterol breakdown) were higher in the extracellular inhibitor group, suggesting that extracellular CTSD inhibition leads to a better metabolic profile in NASH mice [57]. Not only mice demonstrated a beneficial effect on NASH progression after CTSB or CTSD inhibition; additionally, in Sprague-Dawley rats on a high-fat diet (simulating NASH), inhibition of CTSB (through exercise) and CTSD (through compound) improved liver function [58,59]. In summary, CTSB and CTSD seem to play an important role in the progression of pathologies related to the metabolic syndrome, including NASH.

While CTSB and CTSD clearly influence NALFD/NASH progression, CTSS did not seem to have an influence. Even though obese patients and patients who had T2D combined with cardiovascular disease showed increased levels of plasma CTSS, NALFD patients did not have different levels of plasma CTSS compared to healthy controls [60]. Furthermore, genetic variants of *CTSS* need to be taken into consideration since certain genetic variants seemed to be associated with traits related to obesity, such as BMI [61]. However, there is not much information regarding the role of CTSS and CTSL in NAFLD/NASH and in metabolic syndrome development or progression and thus further research should be conducted in this area.

While most of the studies mentioned above regarding cathepsins and metabolism are descriptive, mechanistic studies have demonstrated that the amelioration of NASH-related symptoms upon CTSB inhibition might be linked to the restoration of the levels of the master metabolic regulator sirtuin-1 (SIRT1) [62,63,64], which plays an essential (beneficial) role in the metabolic regulation of lipids, glucose, and insulin [65].

### 3.3. Cathepsins Exacerbate NASH by Stimulating the Immune System

In NASH, the immune system is deregulated and proinflammatory [66]. Cathepsins play a central role in this deregulation and the involvement of various cathepsins has been demonstrated in several studies. In an in vivo model of NASH, mice had upregulated expression of inflammatory genes, such as *TNF-α* and *IFN-γ*, and increased serum levels of proinflammatory cytokines such as IL-1β and IL-18. This gene expression and the serum levels of these proinflammatory cytokines decreased upon administration of a CTSB inhibitor [55,64]. Not only CTSB inhibition decreased proinflammatory markers; additionally, the inhibition of CTSD resulted in decreased expression of the proinflammatory markers *TNF-α*, *CCL2*, and *Caspase 1* [56].

A possible mechanism of action for CTSB to induce a proinflammatory phenotype in NASH is through the abovementioned SIRT1 (see Section 3.2). SIRT1 exerts an inhibitory effect on the inflammatory regulator NFκΒ [67]. Upon inhibition of SIRT1 by CTSB, NFκΒ activates the NLRP3 inflammasome [64,68], leading to increased secretion of the proinflammatory cytokines IL-1β and IL-18. Notably, several cathepsins (e.g., CTSB, CTSS, and CTSL) can also directly activate the NLRP3 inflammasome [69,70,71].

Macrophages play a central role in the pathogenesis of NASH through the creation of a proinflammatory environment by producing various cytokines and chemokines and thereby activating other surrounding (immune) cells [66]. Cathepsins have been shown to play a role in macrophage polarization in NASH. *CTSB* knockout NASH mice (FPC-fed mice) demonstrated higher expression of the anti-inflammatory phenotype macrophage “M2” marker *Arg1* and decreased expression of the proinflammatory phenotype macrophage “M1” markers *iNOS* and *F4/80* compared to NASH mice with normal *CTSB* expression, suggesting that in NASH mice, cathepsins induce proinflammatory M1 macrophages [54].

## 4. Mechanisms of Cathepsins in HCC

Cathepsins are ubiquitously expressed within various tissue types and especially in the context of tumor biology, often secreted into the tumor microenvironment (TME). Many cell types, including macrophages and fibroblasts, contribute to the pool of cathepsins in the TME [3]. Here, we describe the mechanisms by which cathepsins operate in HCC by using the hallmarks of cancer concept [16,17,18]. Hereby, the hallmarks where the involvement of cathepsins is only marginally or not described (evading growth suppressors, enabling replicative immortality, unlocking phenotype plasticity) are left out.

### 4.1. Cathepsins Sustain Proliferative Signaling through the PI3K/Akt/mTOR Signaling Pathway in HCC

Cancer cells have developed the ability to uncontrollably proliferate without the need for external proliferative signals. This ability is modulated by pathways that are involved in cell proliferation and cell growth, thereby moving away from the carefully orchestrated homeostasis of normal cells [17].

A plethora of studies have pointed towards a role for cathepsins in the progression of HCC. In vitro overexpression of *CTSB* and *CTSL* in various HCC cell lines stimulated cell growth and proliferation in at least two independent studies [13,72]. In line, the silencing of *CTSB*, *CTSL*, and *CTSS* in several HCC cells resulted in the reverse, i.e., decreased cell proliferation [13,72,73]. Similarly, it was demonstrated in vivo that the injection of *CTSL*-overexpressing HCC (MHCC-97-H) cells into mice resulted in increased growth and tumor weight compared to the control [13]. Furthermore, when mice were injected with HCC (HepG2) cells that were silenced in *CTSB*, tumor size and weight decreased [72].

The involvement of cathepsins in cancer proliferation is related to its ability to affect intracellular signaling pathways. Several studies demonstrated that cathepsins upregulate the PI3K/Akt/mTOR pathway (crucial in cell growth, proliferation, and survival [74]) through the phosphorylation of the kinase Akt (pAkt) [72,75,76]. In HCC (BEL-7402) cells overexpressing *CTSB*, the levels of pAkt were elevated compared to the control. Likewise, HCC (HepG2) cells with downregulated *CTSB* displayed reduced levels of pAkt [72]. The decrease in pAkt was also observed in acute myeloid leukemia cells, upon silencing of *CTSB* [75]. Other studies demonstrated that mice with breast cancer that had a *CTSD* knockout (MMTV-PyMT-cre; *CTSD*^−/−^) elicited impaired mTOR signaling, which is an important downstream target of pAkt [76], suggesting that cathepsins are involved in this crucial cell survival and growth pathway. Although no direct evidence is available regarding the involvement of CTSS and CTSL in the PI3K/Akt/mTOR pathway in HCC proliferation, the observation that in glioblastoma cells, the inhibition of CTSS reduces the phosphorylation of Akt [77] suggests that the involvement of cathepsins in this pathway is common in different types of cancers, including HCC.

Overall, available data suggest intracellular cathepsins are active players in uncontrolled cancer cell proliferation, specifically by interfering with the pro-survival and proliferation PI3K/Akt/mTOR pathway.

### 4.2. Elevated Levels of CTSB and CTSD Result in Apoptosis Whereas Elevated CTSS Levels Lead to Resistance of Apoptosis in HCC

Tumor cells have acquired the ability to resist programmed cell death or apoptosis and to uncontrollably expand in number. The role of cathepsins in these processes has been described in tumors of different origin, including HCC. While intracellular CTSB and CTSD have been demonstrated to be pro-apoptotic, intracellular CTSS has been found to be anti-apoptotic in HCC (thus, tumor promoting).

The importance of lysosomal disruption and the subsequent release of CTSB for apoptosis in HCC (Huh-7, Hep3B) cells has been elegantly demonstrated by Guicciardi et al. [78]. Their data show that apoptosis is triggered by the TNF-related apoptosis-inducing ligand (TRAIL) pathway (Figure 1). Activation of this pathway eventually leads to the destabilization of the lysosome and to the lysosomal release of CTSB [78]. In addition, TNF-α combined with cycloheximide increased the concentration of sphingosine inside the lysosome, which in turn triggered LMP in rat (HTC), murine (1c1c7), and human HCC (SK-HEP-1) cells, resulting in the release of CTSB and CTSD [79,80]. In cancers, including in HCC, this pathway is often prevented by the upregulation of the anti-apoptotic protein cFLIP (cellular FLICE/caspase-8 inhibitory protein), since this stabilizes the lysosome [78].

Strikingly, permeabilization of the lysosomal membrane in HCC cells resulted in the release of CTSB and CTSD into the cytosol of HCC cells, where they then cleaved essential apoptotic substrates [78,79,81], including the pro-apoptotic Bid, which then turns into its truncated version, tBid [42,79,82,83]. In addition, it was determined that upon lysosomal CTSD release in HCC (BEL-7404) cells, the pro-apoptotic protein Bax was inserted into the mitochondrial membrane [83,84]. tBid formation and mitochondrial Bax insertion result in the release of mitochondrial cytochrome c and subsequently caspase activation, which leads to apoptosis [82].

While the abovementioned studies described a role for CTSB in HCC, CTSB also seems to play a role in non-cancerous liver cells. In non-cancerous rat hepatocytes, CTSB inhibition reduced apoptosis activity [85]. In addition, cells in *CTSB*^−/−^ mice were resilient to TNF-α-induced apoptosis, suggesting the importance of this protease in apoptosis [86].

In contrast to CTSD and CTSB stimulating apoptosis, CTSS actually inhibits apoptosis and thereby promotes tumor progression. *CTSS* is usually highly expressed in immune cells but also in malignant cells, including HCC cells [4]. *CTSS* knockdown in HCC (MHCC-97-H) cells resulted in higher concentrations of cleaved caspase 3 and increased apoptosis [73,87]. Similar effects on the inhibition of CTSS-induced apoptosis were observed in human glioblastoma and renal carcinoma cells [77,88,89]. Hence, these results suggest that CTSS appears to be crucial in preventing apoptosis and that targeting it may be an interesting approach to tackle this evasion of apoptosis in cancer cells.

While there is no information available on the role of CTSL in apoptosis in HCC, mice with pancreatic islet cancer with a knockout in *CTSL* demonstrated a 337% increase in apoptosis [90]. In line, the injection of lung cancer cells with decreased *CTSL* expression in mice showed increased apoptosis in tumors compared to mice injected with tumor cells with normal *CTSL* expression [91]. Hence, the role of CTSL seems to be similar to CTSS, i.e., to inhibit apoptosis.

Altogether, cathepsins play a key role in modulating apoptosis in multiple types of cancer. The specific type of cathepsin and possibly the type of cancer determines whether this role is pro- or anti-apoptotic.

### 4.3. Cathepsins Induce Angiogenesis in HCC

Angiogenesis is the process whereby new blood vessels form from existing vessels. This process occurs in physiological conditions via tightly regulated mechanisms [92]. Whereas, in tumors, this process is severely deregulated. Tumors rely on new blood vessels for continuous growth and tumor cells can self-induce angiogenesis (angiogenic switch) by secreting pro-angiogenic factors such as vascular endothelial growth factor (VEGF) and basic fibroblast growth factor (bFGF).

HCC is characterized by hypervascularity with vascular structural and functional abnormalities [93,94]. This importance of angiogenesis in HCC is also reflected by the fact that most approved treatments for advanced HCC patients, including sorafenib, target several angiogenic pathways, including the pro-angiogenic VEGF/VEGF receptor (VEGFR) pathway [40,93]. In addition, the serum levels of VEGF have been shown to be increased along the progression of HCC. Furthermore, HCC patients with high serum levels of VEGF also showed a significant lower overall survival compared to patients with low serum levels of VEGF (±8 vs. 38 months [95]) [96]. In addition, the pro-angiogenic bFGF was found to be increased in HCC and a better biomarker for HCC diagnosis compared to the standard marker AFP (based on AUROC curves) [95].

In HCC, predominantly CTSB and CTSS have been demonstrated to be positively associated with angiogenesis in HCC (Figure 2) [97]. Cathepsins seem to be specifically involved in the degradation of the extracellular matrix (ECM), an essential process to allow new vessel sprouts to arise. Several in vitro studies demonstrated the effect of cathepsin inhibition on angiogenesis. Upon *CTSB* knockdown in HCC (Huh-7 cells transfected with hepatitis B spliced protein) cells, specific proteins involved in angiogenesis (matrix metalloprotease, MMP; and urokinase-type plasminogen activator, uPA) were reduced in gene expression. In addition, when the medium of these cells treated with siRNA for *CTSB* was transferred to endothelial cells (HMEC-1 cells), they observed reduced endothelial capillary formation compared to a control [98]. In addition to CTSB, CTSS has also been demonstrated to be involved in angiogenesis in HCC. Upon knockdown of *CTSS* in HCC cells (MHCC-97-H cells), reduced secretion of pro-angiogenic factor VEGF was observed. Furthermore, endothelial (HUVEC) cells showed reduced tube formation after being treated with medium obtained from the siRNA-treated HCC cells [73,99]. In line, *CTSS* overexpression in endothelial (HUVEC) cells increased tube formation compared to the control [99].

Other cathepsins might also play an important role in the angiogenesis process. Although not directly demonstrated for HCC, studies using other types of cancer demonstrated a role for both CTSL and CTSD in angiogenesis.

In a co-culture of *CTSL*-knocked-down gastric cancer cells with endothelial cells, a reduction in tubule formation was observed whereas *CTSL* overexpression led to increased tubule formation. Furthermore, *CTSL* knockdown in vivo through the use of the chorioallantoic membrane (CAM) model decreased the number of blood vessels around the gastric cancer tumor compared to the control [100,101]. Taking into account the observed overexpression of *CTSL* in HCC and its function in ECM degradation, these data in gastric cancer suggest that CTSL also stimulates angiogenesis in HCC. In line, mice xenografted with breast cancer cells that were overexpressed in *CTSD* demonstrated an increase in the number of tumor micro-vessels compared to mice that were xenografted with cells with normal *CTSD* expression [102]. Similar to *CTSL*, *CTSD* is overexpressed in HCC and can cleave the ECM. Therefore, it is suggested that CTSD also plays an important role in HCC angiogenesis.

### 4.4. Cathepsins Play a Crucial Role in Activating Invasion and Metastasis in HCC

Most cancer-related deaths occur due to infestations of the cancer cells into different body parts, or metastases. Cancer cells can leave their primary tumor location through the blood vessels to invade distant body sites. By invading, the malignant cells have to cross the basal membrane and the interstitial connective tissue [18]. Proteases play an essential role in these processes [103].

The role of cathepsins in the process of invasion and metastasis is illustrated in Figure 3 [104,105]. They are involved in three stages of tumor invasion: (1) cathepsins that migrate to the cell membrane have the ability to activate other ECM-degrading proteases, such as MMPs and uPAs; (2) cathepsins that are secreted by cancer or by immune cells in the TME can cleave (together with other proteases) the ECM or the basement membrane; and (3) cathepsins can break the interaction between cells, by cleaving the adherens junction components, such as E-cadherin [105]. Cathepsins can cleave these targets of tumor invasion upon attachment to the cancer cell membrane or they can be secreted into the extracellular milieu. CTSB, for example, is often found at the cell membrane in malignant cells and at the edge of the invasive tumor [103,106,107].

In HCC, CTSB induced the invasion of a cohesive multicellular group of cells (collective invasion) in a 3D invasion in vitro (HepG2 and Huh-7 cells) model. In this study, the expression of *CTSB* was enhanced through CD147, a transmembrane protein that is thought to be involved in invasion and metastasis [108]. The role of CTSB in invasion and migration was further established in wound healing and transwell invasion assays (in Huh-7 cells). In addition, *CTSB* overexpression in HCC cells with low metastatic potential (MHCC-97-L cells) demonstrated increased migration and invasion. In line, when *CTSB* was knocked down in an HCC cell line with high metastatic potential (MHCC-97-H), these characteristics diminished [11]. Furthermore, *CTSB* knockdown led to reduced expression of important proteases: MMP9 and uPA [11,98]. Along the same line, evidence points towards the involvement of CTSS in invasion and metastasis. *CTSS* is barely expressed in normal hepatocytes but significantly expressed in HCC hepatocytes [43]. Knockdown of *CTSS* in HCC (MHCC-97-H) cells with high metastatic potential significantly decreased invasion in a transwell invasion assay and motility in a wound healing assay [73]. Moreover, an increased *CTSS* expression level has been shown to be positively correlated with extrahepatic metastasis in HCC patients [43].

While the role of CTSD and CTSL has not been specifically demonstrated in HCC, CTSD has been shown to be directly involved in invasion and metastasis in various other cancer types, including breast cancer. *CTSD* overexpression in breast cancer (MCF-7 and MDA-MB-231) cells promoted migration and invasion, and in addition, it reduced the protein levels of epithelial markers, such as E-cadherin. Similarly, in this study, *CTSD* overexpression in vivo increased the number of metastases [109]. In line, a role for CTSL in invasion and metastasis was demonstrated in ovarian carcinoma cells. Overexpression of *CTSL* in these cells enhanced their ability to invade and metastasize, whereas downregulation of *CTSL* decreased these abilities [110]. Considering the role of CTSD and CTSL in other cancer types and their overexpression in HCC, these data suggest that these cathepsins also stimulate invasion and metastasis in the context of HCC.

### 4.5. Cathepsins Take Part in the Reprograming of Energy Metabolism in HCC

The energy metabolism in cancer cells is frequently deregulated to sustain their increased energy requirements for continuous growth and proliferation. An example of deregulated energy metabolism is the “Warburg effect”, which entails that cancer cells prefer glycolysis over oxidative phosphorylation [17]. It became clear recently that “metabolic reprogramming” is imperative for the cancer to prevail and progress. Several metabolic pathways, such as the glutaminolysis pathway that catabolizes glutamine into the metabolites glutamate or α-ketoglutarate, are increased in cancer [111]. Furthermore, several metabolic factors, such as insulin-like growth factor 1 (IGF-1), can influence growth and survival pathways, e.g., the PI3K/Akt/mTOR pathway [112].

Cathepsins play a role in this metabolic deregulation through IGF-1 signaling. IGF-1 has been shown to significantly promote cell proliferation, migration, and invasion in HCC (Hepa1-6 and H22) cell lines [113]. This pro-cancerous effect was exerted through CTSB, as upon CTSB depletion, the IGF-1 stimulatory effects were reversed. Furthermore, when WT mice and diabetic (KK-ay; with high IGF-1 levels) mice were xenografted with HCC (Hepa1-6) cancer cells, it was observed that the cancer expanded faster in the diabetic mice (i.e., by, among others, the high levels of IGF-1). Furthermore, upon knockdown of *CTSB*, the diabetic mice showed a significant reduction in tumor expansion compared to the diabetic mice that had normal CTSB levels, indicating that CTSB was the effector protein of IGF-1 [113]. Similarly, upon *CTSL* knockdown, less proliferation and migration were observed in vitro upon stimulation with IGF-1 compared to cells that had normal *CTSL* expression [91].

Not only IGF-1 is an upstream factor of cathepsin activity. Additionally, the addition of glucose led to increased levels of cathepsins in vitro. Sung et al. demonstrated that the addition of glucose to the medium of human peritoneal mesothelial cells led to an increase in the CTSB levels [114]. Since glucose metabolism is upregulated in HCC [115], this points towards a potential link between deregulated metabolism and cathepsin activity.

### 4.6. Cathepsins Promote Tumor Immune Evasion

Immune surveillance is a process whereby immune cells monitor to detect and eliminate cancer cells. Cancer cells have, however, developed a mechanism to avoid this immune destruction [116]. Malignant cells can secrete immunosuppressive factors such as TGF-β, which recruits regulatory T cells, myeloid-derived suppressor cells (MDSCs), and other immunosuppressive cells, leading to the suppression of the immune response [17].

Cathepsins can be expressed and secreted by immune cells and they have physiological functions in the immune system [117]. Cathepsins have been found to play a part in toll-like receptor signaling and cytokine activation/inhibition [118,119]. In addition, cathepsins, mainly CTSS, play an important role in the processing and presenting of antigens in the antigen-presenting cells, e.g., macrophages and dendritic cells [120].

Cathepsins not only have a physiological function in the immune system but are also key players in the evasion of the immune system present in cancers. The TME is a significant contributor to the suppressed immune response that is present in HCC [121]. Among the cells present in the TME, including in HCC, tumor-associated macrophages (TAMs) are key players. TAMs have an M2 macrophage-like anti-inflammatory phenotype, unlike the M1 macrophage phenotype, which is proinflammatory [122]. Because TAMs suppress the immune response, they are tumor-promoting; hence, their presence is generally related with poor prognosis [123]. TAMs secrete CTSB, CTSL, and CTSS [124,125] and therefore, TAMs also provoke tumor growth, angiogenesis, invasion, and metastasis responses. Inhibition of these cathepsins in vitro (human M2 polarized macrophages derived from PBMCs used as a representative cell of TAM) led to an increase in the expression of proinflammatory cytokines such as *TNF-α* and *IL-1β* and also to increased protein levels of NOS2 (M1 marker), among others [125]. As such, these data suggest that CTSB, CTSL, and CTSS inhibition in M2 macrophages drives a shift to M1 polarization [125,126]. Hence, the expression of *CTSB*, *CTSL*, and *CTSS* represents an important way for TAMs to evade the immune system. In line, other myeloid cells in the TME have been shown to gain an M2 phenotype, mainly through CTSS. These M2 phenotype cells favor the immunosuppressive regulatory T cells over cytotoxic CD8^+^ T cells, contributing even further to the immunosuppressive environment [127]. In addition, the TME contains MDSCs, which also harbor an immunosuppressive function. Studies found that the expression of *CTSB* was increased in MDSCs, thereby contributing to HCC progression as well [128].

This immunosuppressive-stimulating phenotype driven by cathepsins is contrary to the data observed in NASH, where CTSB induced the macrophage polarization from the anti-inflammatory M2 to the proinflammatory M1 macrophages. This shift in macrophage polarity can possibly point towards a transitory role for cathepsins from NASH to HCC through the immune system. The importance of the immune system in the NASH to HCC transition is also demonstrated in NASH-HCC because immune deregulation is an essential part of the NASH-HCC disease phenotype. Patients with NASH-HCC are less responsive to immune therapy [129]. This phenomenon was also demonstrated in preclinical animal studies since NASH-HCC mice that were treated with anti-programmed death-1 (PD1)-induced CD8^+^ T cells did not have tumor regression but rather increased the number and size of the tumor lesions. As CD8^+^ T cells are generally believed to induce an anti-tumor response, this finding was surprising: it was suggested that the CD8^+^ T cell tumor immune surveillance was impaired through the exhaustion of these cells in NASH-HCC and that these exhausted CD8^+^ T cells, instead of preventing it, contributed to the progression of HCC from NASH [129,130]. Since cathepsins play such a big role in the immune deregulation in both NASH and HCC, this observation is suggestive of cathepsins being key drivers of such deregulation in NASH-HCC.

Besides the cells in the tumor microenvironment, the HCC cells can also contribute to the immunosuppressive phenotype through the previously described SIRT1 (see Section 3.2 and Section 3.3). In contrast to NASH, where SIRT1 levels are low due to the inhibitory effect of cathepsins [64], the levels of SIRT1 in HCC cells are high and seem to have a pro-tumor role by inducing an immunosuppressive phenotype, among others [131,132,133,134].

Hence, these data suggest that, in general, cathepsins play an active role in the deregulated immune system surrounding HCC and that they might be involved in the transition from NASH to HCC through this process.

### 4.7. Genomic Variation in Cathepsin Genes Can Alter Tumor Development

Genetic alternations are the driving force behind cancer development. Several single nucleotide polymorphisms (SNPs) in the genes that encode cathepsins have been associated with cancer development. Chen et al. demonstrated that in an Asian population, an SNP present in the *CTSB* gene (rs13332) was significantly associated with the risk of HCC (with an adjusted odds ratio of 2.2) [135]. In addition, Cui et al. showed that the genotype and allele distribution of a *CTSB* SNP (rs12898) between HCC patients and healthy controls was significantly different [136].

For CTSL and CTSD, there is no direct evidence available for the effect of genetic variations in HCC or cancer in general. *CTSS*, however, is often mutated in follicular lymphoma. It was determined that a specific variation (Y132D) increases CTSS activity, which has tumor-promoting effects [137].

These data indicate that variations in the cathepsin genes can promote cancer and that more research should be carried out to study the effect of the various existing variants.

## 5. Therapeutic Potential of Cathepsins in NASH-HCC

Since treatment options are limited for NASH-HCC, the need for new therapeutic targets is high. A growing body of evidence points towards the involvement of cathepsins in multiple processes related to the pathology of NASH and hallmarks of cancer, suggesting that cathepsins can be used as a promising diagnostic tool and therapeutic target for NASH-HCC. In line with this view, several therapeutics that target the activity of cathepsins directly are currently tested in vitro, in preclinical models and in clinical trials. Due to the acidic environment in cancers, which is essential for their activation, cathepsins can also be used for targeted drug delivery, where a prodrug is delivered to the cancer site and activated through cathepsin cleavage [9]. One example of this approach is ADCETRIS^®^, an antibody–drug conjugate that uses the cleaving activity of CTSB and, since 2021, has been in clinical use as a treatment for several types of lymphoma [138,139].

However, before considering modulation of cathepsins activity as a therapy, several challenges need to be overcome. First, cathepsins are not specifically involved in single pathways but are rather involved in multiple complex pathways, including pathways that are essential for physiological functions [3]. Second, multiple cathepsins can be involved in the same pathway and several cathepsins can also regulate exocytosis of other cathepsins [140], suggesting a compensatory mechanism or broad side effects upon inhibition of specific cathepsins. Third, the (sub)cellular localization of cathepsins is crucial to their function and the activity of cathepsins could also be dependent on the type and/or stage of the disease [141,142] and variation in (epi)genetics, diet, or other lifestyle factors that have been shown to alter cathepsins functioning [135,143,144]. Another way cathepsins could be utilized as a therapeutic target is by focusing on their relevant downstream effectors, such as SIRT1. Targeting these downstream effectors represents a more directed approach, which possibly prevents some of the hurdles described above regarding direct cathepsin inhibition.

Furthermore, despite multiple studies demonstrating the association between plasma cathepsins levels and NASH and HCC, cathepsins are still not used as diagnostic tools to predict early HCC in NASH patients. Among the reasons for the absence of progress in this aspect are the lack of data in which measurements of cathepsins were performed before and after the onset of HCC in the same patients, variation between different cohorts, and the lack of knowledge regarding the daily triggers of exocytosis of cathepsins in health and disease.

Thus, to maximize the clinical utilization of cathepsins, studies determining their specific regulation under multiple conditions and in different localizations and exploring the upstream and downstream effectors that are involved in their regulation should be performed.

## 6. Conclusions

Cathepsins are important players in the disease progression of both NASH and HCC. They are actively involved in many driver pathways of pathology and are expressed by a variety of cells in the tumor microenvironment. The pathways by which cathepsins contribute to NASH are apoptosis, metabolism, and induction of the immune system. Key cancer progression pathways in which cathepsins are involved include proliferation, apoptosis, increased invasion, metastasis and angiogenesis, deregulated energy metabolism, and evasion of the immune system (Figure 4). Therefore, it is likely that the involvement of cathepsins in NASH-HCC is greater than HCC derived from other etiologies. Since the immune system deregulation changes from proinflammatory to anti-inflammatory and NASH-HCC patients respond worse to immune checkpoint inhibition therapy, it seems that this is important in the transition from NASH to HCC. Cathepsins play an important role in this deregulation. As cathepsins have versatile pro-disease functions, they form a viable therapeutic target in NASH-HCC, a condition in which therapeutic options are limited.

## Figures and Tables

**Figure 1 biomedicines-10-02351-f001:**
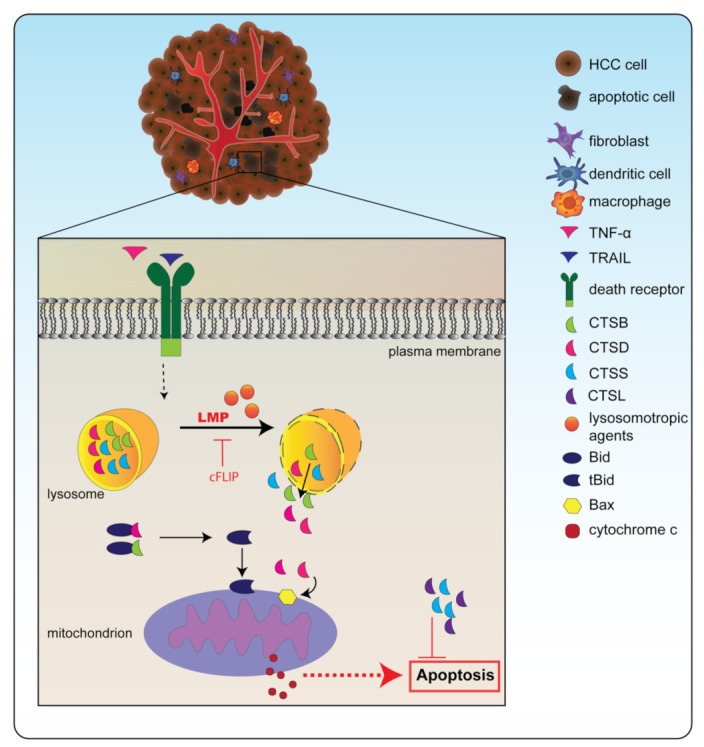
Cathepsins are involved in apoptosis. Destabilization of the lysosome through factors such as TNF-related apoptosis-inducing ligand (TRAIL) or tumor-necrosis factor α (TNF-α) combined with a lysosomotropic agent leads to lysosomal membrane permeabilization (LMP). Through LMP, cathepsins can be released from the lysosome into the cytosol. In the cytosol, cathepsin B (CTSB) and cathepsin D (CTSD) can cleave pro-apoptotic factor Bid into its active version truncated (t)Bid. In addition, these cathepsins can stimulate Bax insertion into the mitochondrial membrane. Together, this leads to the release of cytochrome c, which can induce the caspase cascade, eventually culminating in apoptosis. Cathepsin S (CTSS) and cathepsin L (CTSL), however, can inhibit apoptosis and therefore when they are upregulated, cancer cells can resist this form of cell death. Note: the mechanisms of cathepsin-induced apoptosis are most likely similar to the mechanisms occurring in NASH.

**Figure 2 biomedicines-10-02351-f002:**
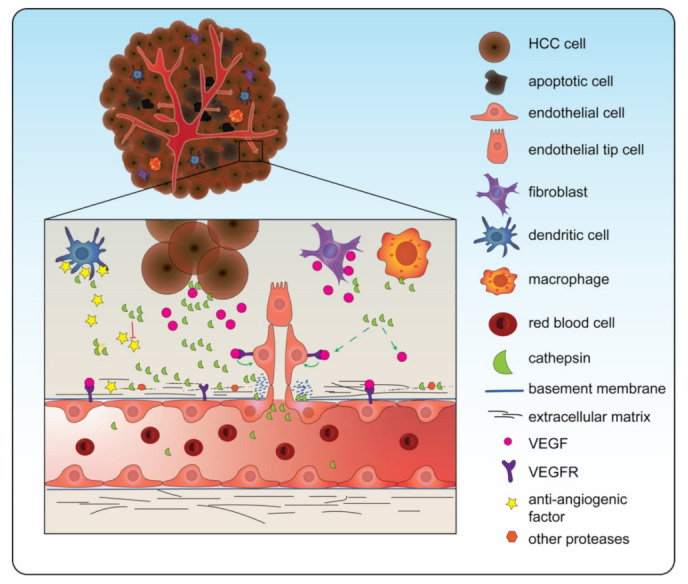
Cathepsins are involved in angiogenesis. Cathepsins are secreted into the tumor microenvironment (TME) by cancer cells but also by other cells from the TME such as fibroblasts and macrophages. These extracellular cathepsins increase the levels of vascular endothelial growth factor (VEGF), which induces angiogenesis. Furthermore, they cleave the extracellular matrix components, making it possible for the growing tip to develop. In line, cathepsins cleave anti-angiogenic factors. Note: in this figure CTSD, CTSB, CTSS, and CTSL are all depicted as one green symbol for ease of concept.

**Figure 3 biomedicines-10-02351-f003:**
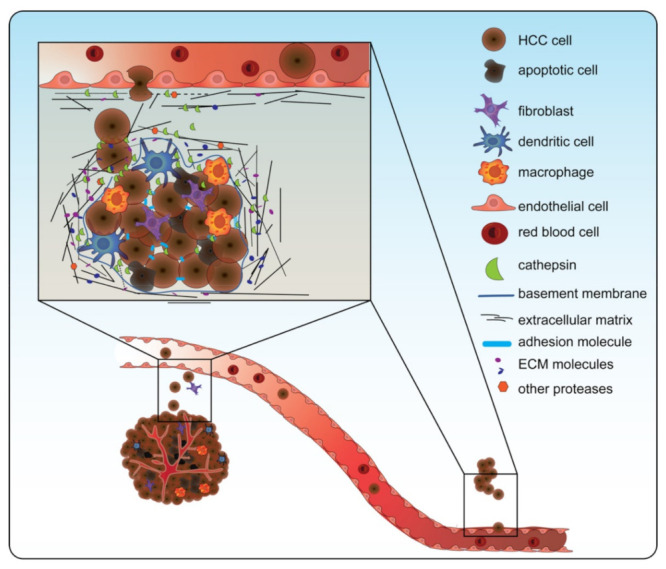
Cathepsins are involved in tumor invasion and metastasis. Cathepsins are involved in tumor invasion and metastasis by (1) going to the surface of the cancer cell or being secreted to activate other proteases such as matrix metalloproteases (MMPs) and urokinase-type plasminogen activator (uPAs); (2) cleaving the extracellular matrix (ECM) together with the other proteases; and (3) breaking the junctions between cells. In doing so, cathepsins contribute to the epithelial-to-mesenchymal transition, aiding in invasion through the ECM and entry into the blood vessel. Note: in this figure, CTSD, CTSB, CTSS, and CTSL are all depicted as one green symbol for ease of concept.

**Figure 4 biomedicines-10-02351-f004:**
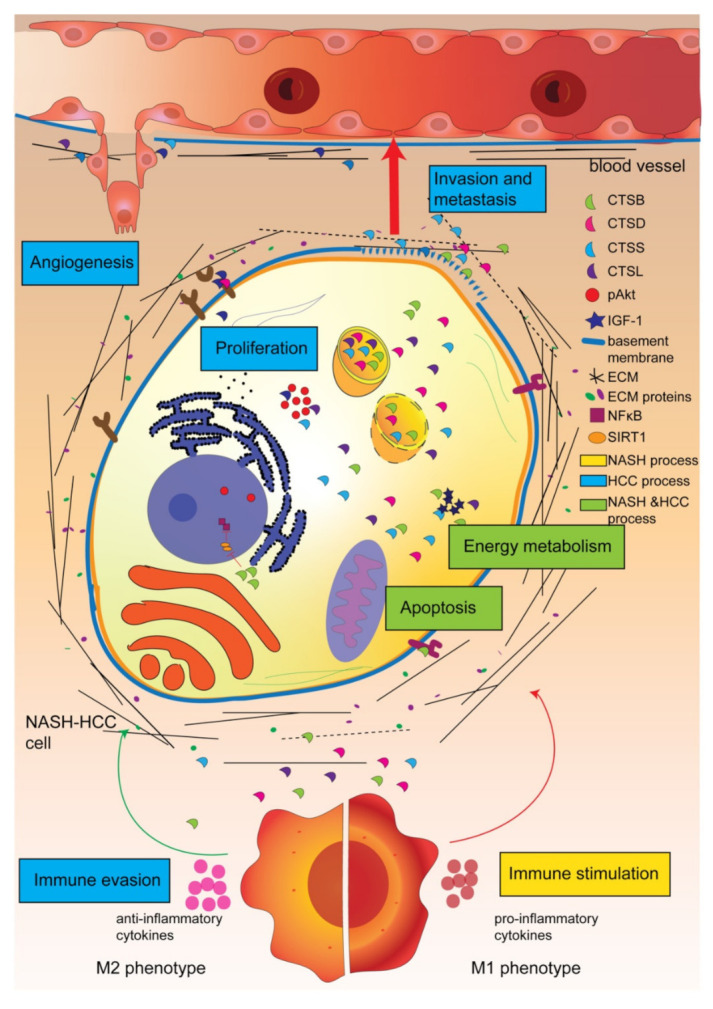
Conclusion and hypothesized working mechanism of cathepsins in NASH-induced hepatocellular carcinoma (HCC). Cathepsins are involved in NASH through actions in (1) apoptosis, (2) energy metabolism, and (3) immune stimulation, and they are involved in HCC through actions in (1) proliferation, (2) apoptosis, (3) angiogenesis, (4) invasion and metastasis, (5) deregulation of energy metabolism, and (6) evasion of the immune system. Cathepsins help to stimulate the immune system in NASH whereas they suppress the immune system in HCC.

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
