# Peer review of "All Roads Lead to Cathepsins: The Role of Cathepsins in Non-Alcoholic Steatohepatitis-Induced Hepatocellular Carcinoma"

_biomedicines, 2022, doi:10.3390/biomedicines10102351_

Round 1
Reviewer 1 Report
The article "All roads lead to cathepsins –The role of cathepsins in non-alcoholic steatohepatitis induced hepatocellular carcinoma" was written based on the cellular and immunological remodeling role of cathepsins toward HCC. I appreciate the author's detailed collection sequence and clarity. I have a few suggestions
1. Authors could be explained the severity of NAFLD and NASH related to/leading to HCC.
2. Genetic alteration mechanism by cathepsins of HCC may be included in more detail.
Author Response
Point 1: Authors could be explained the severity of NAFLD and NASH related to/leading to HCC.
Response 1: We would like to thank the reviewer for his/her positive words regarding our manuscript and acknowledging the relevance of our work. We agree that the severity of NAFLD and NASH related to HCC could be further addressed. In the revised manuscript we therefore adjusted this as follows:
“NAFLD patients have an increased risk to develop liver-related morbidity and mortality [23]. In addition, NAFLD patients are at high at risk for developing cardiovascular disease, chronic kidney disease and extrahepatic malignancies (e.g., colorectal, uterine and breast cancer) among others [24,25]]” (See chapter 1, p.2, line: 70-73).
Furthermore, we added data regarding the prevalence of the development of cirrhosis and HCC from NASH.
“In addition, the liver of these patients can become fibrotic, posing them at risk for the development of liver cirrhosis (5-18% for NASH patients without fibrosis, up to 38% for NASH patients with fibrosis [26]) and hepatocellular carcinoma (HCC) (2.4% - 12.8% for cirrhotic NASH patients [27]) [28,29].” (See chapter 1, p.2, line: 76-80)
Point 2: Genetic alteration mechanism by cathepsins of HCC may be included in more detail.
Response 2: We thank the reviewer for his useful suggestion. In the revised version, we added a paragraph to address this comment.
“Chapter 4.7: Genomic variation in cathepsin genes can alter tumor development.
Genetic alternations are the driving force behind cancer development. Several single nucleotide polymorphisms (SNPs) in the genes that encode cathepsins have been associated with cancer development. Chen et al. demonstrated that in an Asian population a SNP present in the CTSB gene (rs13332) was significantly associated to the risk of HCC (with an adjusted odds ratio of 2.2) [135]. In addition, Cui et al showed that the genotype and allele distribution of a CTSB SNP (rs12898) between HCC patients and healthy controls was significantly different [136]. For CTSL and CTSD there is no direct evidence available for the effect of genetic variations in HCC or cancer in general. CTSS, however, is often mutated in follicular lymphoma. It was determined that a specific variation (Y132D) increases CTSS activity, which has tumor promoting effects [137]. These data indicate that variations in the cathepsin genes can promote cancer and that more research should be done to study the effect of the various existing variants.” (See chapter 4.7 p. 12, line: 509-522).
Furthermore 1 line was added regarding this topic for obesity:
“Furthermore, genetic variants of CTSS need to be taken into consideration since certain genetic variants seemed to be associated with traits related to obesity, such as BMI [61]”
(See in chapter 3.2, p.4, line: 175-176)
Reviewer 2 Report
In their manuscript "All roads lead to cathepsins – The role of cathepsins in non-alcoholic steatohepatitis induced hepatocellular carcinoma" H. van Mourik et al. report on the role of cathepsins in the development of NASH and HCC disease. In general, the review has good storytelling, which includes literature-confirmed statements at in vitro, cellular, and in vivo levels. The authors collected a lot of materials on the role of (overexpressed) cathepsins on the immune system, the mechanism of activation, and the cellular mechanism of apoptosis. Overall, the work is great and can be accepted without any modifications (except for text formatting and picture quality improvements).
Author Response
Point 1: Overall, the work is great and can be accepted without any modifications (except for text formatting and picture quality improvements).
Response 1: We would like to thank the reviewer for his/her positive words regarding our manuscript and for acknowledging the relevance of our work. We will ensure that the text formatting and picture quality will be thoroughly checked in the PDF proofing version.
Reviewer 3 Report
The review article entitled “All roads lead to cathepsins –The role of cathepsins in non-alcoholic steatohepatitis induced hepatocellular carcinoma” by Van Mourik and colleagues elegantly offers updated information on the participation of cysteine cathepsins in physiological processes and specially in NASH-HCC. The review is well written, easy to follow, and perfectly structured. There are, however, some minor aspects that need attention:
-Most of the data linking cathepsins to energy metabolism and inflammation are mostly descriptive and circumstantial, since increases/decreases of Cathepsin are associated to enhanced/decreased phenomena (pro-inflammatory mediators, glucose levels, or so on). Thus, no real mechanism of action is provided to verify/prove this link.
In relation to this, there are some articles linking cathepsins with Sirtuin1 (known regulator of metabolism, and also inhibitor of NFkB activation leading to inflammation) that could provide this missing link, and that merits to be discussed in this scenario.
Moreover, and taking into account the fact that cathepsins are not, to date, a good therapeutic target given their redundancy between cathepsin species, and the fact that are involved in many relevant pathways, the feasibility of targeting other intermediate effectors/substrates, for example SIRT1, should be considered in some instances.
-Similarly, the potential participation of CTSB in the activation of the inflammasome, and consequent release of IL1beta and IL18 should be regarded as well.
Minor comment:
-CSTD misspelled in line 128
Author Response
Point 1: The data linking cathepsins to energy metabolism and inflammation are mostly descriptive and circumstantial, since increases/decreases of Cathepsin are associated to enhanced/decreased phenomena (pro-inflammatory mediators, glucose levels, or so on). Thus, no real mechanism of action is provided to verify/prove this link. In relation to this, there are some articles linking cathepsins with Sirtuin1 (known regulator of metabolism, and also inhibitor of NFκB activation leading to inflammation) that could provide this missing link, and that merits to be discussed in this scenario. Similarly, the potential participation of CTSB in the activation of the inflammasome, and consequent release of IL1beta and IL18 should be regarded as well.
Response 1: We would like to thank the reviewer for his/her positive feedback and for his/her useful suggestions. We agree with the reviewer’s comment that the data linking cathepsins to energy metabolism and inflammation are mainly descriptive, and we have now stressed this point in the paper. Furthermore, we are very happy about the reviewer’s suggestion to refer to the link between cathepsins, Sirtuin1 (SIRT1) and the inflammasome.
To address these comments, the following paragraphs were added:
“While most of the studies mentioned above regarding cathepsins and metabolism are descriptive, mechanistic studies have demonstrated that the amelioration of NASH related symptoms upon CTSB inhibition, might be linked to the restoration of the levels of the master metabolic regulator sirtuin-1 (SIRT1) [62-64], which plays an essential (beneficial) role in the metabolic regulation of lipids, glucose and insulin [65].” (See chapter 3.2, p.4, line: 180-184).
“A possible mechanism of action for CTSB to induce a pro-inflammatory phenotype in NASH, is through the abovementioned SIRT1 (see section 3.2). SIRT1 exerts an inhibitory effect on the inflammatory regulator NFκΒ [67]. Upon inhibition of SIRT1 by CTSB, NFκΒ activates the NLRP3 inflammasome [64,68], leading to increased secretion of the pro-inflammatory cytokines IL-1β and IL-18. Notably, several cathepsins (e.g., CTSB, CTSS and CTSL) can also directly activate the NLRP3 inflammasome [69-71].” (See chapter 3.3, p.4, line: 195-200)
“Besides the cells in the tumor microenvironment, the HCC cells can also contribute to the immunosuppressive phenotype through the previously described SIRT1 (see sections 3.2, 3.3). In contrast to NASH where SIRT1 levels are low due to the inhibitory effect of cathepsins [64], levels of SIRT1 in HCC cells are high and seem to have a pro-tumor role by inducing an immunosuppressive phenotype among others [131-134]. (See chapter 4.6, p.12, line: 500-504)
Point 2: Taking into account the fact that cathepsins are not, to date, a good therapeutic target given their redundancy between cathepsin species, and the fact that are involved in many relevant pathways, the feasibility of targeting other intermediate effectors/substrates, for example SIRT1, should be considered in some instances.
Response 2: We thank the reviewer for his/her useful suggestion. To address this point, we added the following sentences into the section regarding the challenges of using cathepsins as therapeutic option:
“Another way cathepsins could be utilized as therapeutic target, is by focusing on their relevant downstream effectors, such as SIRT1. Targeting these downstream effectors would be a more directed approach, which possibly prevents some of the hurdles described above regarding direct cathepsin inhibition.” (See chapter 5, p. 13, line 548-551)
Point 3: CSTD misspelled in line 128
Response 3: We thank the reviewer for reading the manuscript carefully and for pointing towards this mistake, which we have now corrected (see line 134 now).